# Worsening Rhinosinusitis as a Prognostic Factor for Patients with Nasopharyngeal Carcinoma: A Retrospective Study

**DOI:** 10.3390/biomedicines10123235

**Published:** 2022-12-13

**Authors:** Wei-Chieh Lin, Yu-Hung Kuo, Chuan-Jen Hsu, Hung-Pin Wu, Yuan-Jhen Hsu

**Affiliations:** 1Department of Otolaryngology, Head and Neck Surgery, Taichung Tzu Chi Hospital, Buddhist Tzu Chi Medical Foundation, Taichung 427, Taiwan; 2Department of Research, Taichung Tzu Chi Hospital, Taichung 427, Taiwan; 3Department of Otolaryngology, National Taiwan University Hospital, Taipei 100, Taiwan; 4School of Medicine, Tzu Chi University, Hualien 970, Taiwan

**Keywords:** nasopharyngeal carcinoma, prognosis, rhinosinusitis, Lund–Mackay score

## Abstract

Rhinosinusitis is common in patients with nasopharyngeal carcinoma (NPC). Our study aimed to explore the role of rhinosinusitis severity in NPC prognosis. Medical records and radiologic examinations of 90 patients with NPC at a single medical center from 2009–2016 were retrospectively analyzed. The Lund–Mackay (L–M) score was obtained for each patient before and after 6 months of treatment. Rhinosinusitis diagnosis was based on L–M scores of ≥4. L–M score differences were calculated as pre-treatment rhinosinusitis (PRRS) minus post-treatment rhinosinusitis (PSRS). L–M score difference was sub-grouped into “L–M scores > 0”, “L–M scores = 0”, and “L–M scores < 0”. Clinical staging of our patients based on the *American Joint Committee on Cancer 7th edition* were: stage I in nine, stage II in seventeen, stage III in twenty-two, and stage IV in forty-two patients; twenty-seven (30%) patients had died. PRRS incidence was 34.4%, and PSRS was 36.7%. Median of L–M scores difference was 0 (−2.2). L–M score difference was an independent prognostic factor for the overall survival of patients with NPC (*p* < 0.05). Therefore, worsening rhinosinusitis was a prognostic factor for patients with NPC. Clinicians should consider NPC as a warning sign of poor prognosis during routine follow-ups.

## 1. Introduction

Nasopharyngeal carcinoma (NPC) [1] is endemic to Taiwan and Southeast Asia. Approximately 5–10 new cases are reported per 100,000 people yearly in Taiwan [2]. Its incidence peaks in the fifth and sixth decades of life. The male-to-female ratio was 2–3:1. NPC is a distinct head and neck cancer entity due to its multifactorial pathogenesis, including genetic and environmental factors and pathogenic infections, like Epstein–Barr virus (EBV). A theory of “NPC ecology” has been proposed to build a new comprehensive framework of NPC tumorigenesis and tumor progression [3]. The mainstay of NPC treatment was radiotherapy (RT) with or without systemic therapy, which evolved from three-dimensional conformal RT to intensity-modulated RT (IMRT) and volumetric-modulated arc therapy (VMAT) [4,5]. Rhinosinusitis is commonly seen in patients with NPC before and after treatment. Furthermore, 45.2% of patients with NPC in Taiwan are affected with rhinosinusitis before NPC radiation therapy [6]. This phenomenon was best explained by the association between inflammation and carcinogenesis, such as ulcerative colitis for colon cancer [7,8]. Sinonasal tract inflammation was proposed as a precursor of nasopharyngeal carcinoma in a systemic review and meta-analysis in Taiwan and a population-based study in the USA [9,10]. Hung et al. demonstrated that patients with rhinosinusitis had higher risks of NPC than those without rhinosinusitis in Taiwan [11]. In another study about comprehensive risk score and NPC, the author pointed out post-radiation rhinosinusitis as an independent prognostic factor of local recurrence, distant metastasis, and disease-free survival (DFS) [12], suggesting that the inflammatory process of the naso-sinus affected NPC prognosis. There were two unanswered questions about rhinosinusitis and NPC. First, computed tomography (CT) and magnetic resonance imaging (MRI) were the diagnostic tools for rhinosinusitis [13,14,15]. However, in the previous studies addressing the relationship between rhinosinusitis and NPC, there was a lack of qualitative tools for rhinosinusitis diagnosis and severity [11,12]. Second, the relationship between rhinosinusitis severity and NPC prognosis has not been studied yet. The Lund–Mackay (L–M) score system is a tool that radiologically studies rhinosinusitis and has been proven to be well-correlated with severity of rhinosinusitis clinically [15,16]. Therefore, our study aimed to qualitatively investigate the role of rhinosinusitis severity in NPC prognosis using the L–M score system.

## 2. Materials and Methods

### 2.1. Patient Characteristics

Between January 2009 and December 2016, we retrospectively reviewed the medical records and radiological findings of 90 patients who were histologically diagnosed with NPC at a single medical center. Each patient was assessed before the treatment, including physical examination, nasal endoscopy, CT or MRI, whole-body bone scan, and positive emission tomography (PET). The patients with NPC were staged or re-staged according to the 7th edition of *American Joint Committee on Cancer (AJCC)* system. All the patients were diagnosed with NPC without a history of head and neck cancer. They completed the treatment course at our hospital. The treatment modality of each patient was determined by clinicians’ consensus based on the Taichung Tzu Chi Hospital tumor board. This study was approved by the institutional review board of the Taichung Tzu Chi Hospital (protocol code: REC111-65).

### 2.2. Radiation Therapy Protocol

Patients received RT in the supine position with a head and neck thermoplastic mask before treatment. The pre-treatment CT imaging simulation protocol was non-contrast-enhanced 2.5 mm slices from the top of the head to the carina level. Our radio-oncologist analyzed the images, and a Varian Eclipse version 13.6 treatment planning system was set to begin a VMAT plan. The target volumes were delineated according to our institutional treatment protocol, in agreement with the *International Commission on Radiation Units and Measurements* reports 50 and 62. Simultaneous integrated boost (SIB) was used on our patients. Subsequently, 70–74.8 Gy at 2–2.2 Gy/fraction of the planning target volume (PTV) of the gross primary tumor volume (GTV-P) and metastatic lymph nodes > 1 cm (GTV-N) were administered. In contrast, 63 Gy at 1.8 Gy/fraction delivered to the PTV of the microscopic involvement regions, including the nasopharynx, retropharyngeal space, pterygopalatine fossa, pterygoid fossa, maxillary sinus, sphenoid sinus, skull base, clivus, lymph nodes over level II, III, IV, and retropharyngeal lymph nodes (CTV-1) was prescribed. Furthermore, 56 Gy at 1.6 Gy/fraction to the PTV of the low-risk regions (CTV-2) was administered. Over five fractions of this treatment protocol were performed once daily every week. According to the Radiation Therapy Oncology Group standard, limits were placed on the maximum-tolerated dose to decrease the number of organs at risk (OARs).

### 2.3. Chemotherapy Protocol

In advanced NPC (stage II–IVB), 6–7 cycles of cisplatin-based chemotherapy were administered. Cisplatin 30 mg/m^2^ was administered weekly. In addition, neoadjuvant chemotherapy was administered in patients with stage III and IV diseases. The systemic therapy was cisplatin 50–75 mg/m^2^ administration on day 1 and 5-fluorouracil 600–1000 mg/m^2^ on days 1–4 for 2–3 cycles every three weeks. Following RT, chemoradiotherapy (CCRT) was performed.

### 2.4. Follow-Up

After completing the treatment, the patients were followed up in our outpatient clinic at a 1-month interval for the first year, a 3-month interval for 2–3 years, and a 6-month interval yearly. Physical examination and nasal endoscopy were performed at each visit. The treatment-related complications were recorded in medical charts. MRI was scheduled for 3 months after treatment and in the 3-month interval during the first year. NPC treatment response was divided into three groups: complete response (CR), partial response (PR), and stable/progression (residual). CR was 100% remission of the primary tumor observed in imaging or endoscopy. PR was >50% remission of the primary tumor during follow-up. Residual response was when no difference was observed in the primary tumor progression. Sinusitis diagnosis was based on the L–M score system of the CT/MRI studies. The following appearances in CT/MRI were regarded as abnormal:Mucosal thickening of the sinus wall;Air–fluid level in the sinus cavity;Opacification at the ostiomeatal complex;Enhanced and thickened mucosa in the contrast-enhanced study.

The L–M score, proposed by Valerie J. Lund and Ian S. Mackay [17], recorded changes in the frontal, anterior ethmoid, posterior ethmoid, and sphenoid sinuses and ostiomeatal complex. Normal mucosal thickening was scored 0, <50% mucosal thickening was scored 1, and >50% sinus opacification was scored 2. Bilateral paranasal sinuses were recorded separately. Ostiomeatal complex was assigned with a score of “0” or “2”. The total L–M score on each side was 12 points. Furthermore, rhinosinusitis was diagnosed when L–M scores ≥ 4. The L–M scores of pre-treatment sinusitis (PRRS) and post-treatment sinusitis (PSRS) were calculated. L–M scores of PRRSminus L–M scores of PSRS were designated “L–M scores difference.” AllL–M scores were determined by two otolaryngologists (Y.H. and W.L.) in consensus.

### 2.5. Statistical Analysis

Patient characteristics, including age, sex, AJCC stages, treatment modalities, treatment response, follow-up, and sinusitis severity, were expressed as the mean and standard deviation (SD), median and interquartile, or numbers and percentages. Independent-samples Mann–Whitney U-test and independent-samples Kruskal–Wallis test were examined for the differences in continuous variables. Survival rates were illustrated using Kaplan–Meier curves. The log-rank test was introduced for statistical verification of differences. The Cox proportional hazards model revealed crude and adjusted hazard ratios with 95% confidence intervals (CIs). Statistical significance was set at 0.05. Statistical data analysis was performed using Statistical Package for the Social Sciences (SPSS) for Windows (version 22.0, SPSS Inc., Chicago, IL, USA).

## 3. Results

### 3.1. Patient and Disease Characteristics

Ninety patients pathologically diagnosed with NPC in 2009–2016 were analyzed. The mean age was 55.3 ± 12.1 years. Sixty-three patients were male, and twenty-seven were female. According to the *7th AJCC* staging classification, there were nine patients (10%) in stage I, seventeen(18.9%) in stage II, twenty-two in stage III (24.4%), and forty-two(46.7%) in stage IV. Twelve patients received RT only, sixty-six received CCRT, and twelve received neoadjuvant chemotherapy following CCRT (NACCRT). Based on the WHO pathological classification, there was one patient (1.1%) in type I, sixty-nine patients (76.7%) in type II, and twenty patients (22%) in type III. Complete remission was achieved in seventy-one patients (78.9%), partial remission in twelve (13.3%), and seven (7.8%) had residual diseases. Twenty patients (22.2%) developed local recurrence, ten patients (11.1%) developed regional recurrence, and twenty patients (22.2%) developed distant metastasis after treatment. Thirty-one patients (34.4%) had PRRS, and the median L–M score was 2 (0,5), whereas thirty-three patients (36.7%) had PSRS, and the median L–M score was 0 (−2.2). Additionally, the mean follow-up period was 44.9 ± 16.8 months. The patient and disease characteristics are summarized in Table 1.

### 3.2. Factors Associated with the Prognosis of NPC Survival

In the multivariate analysis—T4 stage, PSRS, L–M score difference < 0, and recurrence and residual diseases—were the poor prognostic factors. Furthermore, there was a positive trend between survival and AJCC staging, distant metastasis, and age. Notably, the hazard ratio (HR) of the L–M score difference < 0 was higher than that of the other two groups, which no study reported. HR of different factors on survival are presented in Table 2.

Studies have revealed that the advanced tumor stage plays a prognostic role in NPC and might be a confounding factor for the L–M score difference. Therefore, we further examined the relationship between the L–M score difference and the following variables: tumor stage, nodal status, distant metastasis, AJCC staging, and treatment modalities. Furthermore, no significant association was observed between the Lund score difference and clinical tumor status, as seen in Table 3. Therefore, we conclude that the L–M score difference was an independent prognostic factor in the overall survival of NPC.

### 3.3. The Role of Rhinosinusitis before and after Treatment in the Overall Survival of Patients with NPC

Nine patients with PRRS and seven with PSRS died during follow-up. However, if we adjust the diagnostic criteria to a study by Huang et al. [12], thirteen patients with PRRS and twenty-two with PSRS died. The L–M score difference was obtained from the L–M score of PRRS minus that of PSRS. We further divided the L–M score difference into three groups: L–M score difference “>0”, “=0”, and “<0”. The Kaplan–Meir survival revealed that the three L–M score groups had different progression-free survival (PFS) and overall survival (OS) scores.

Patients with L–M score difference <0 had the worst progression-free survival and overall survival scores (Figure 1a,b). A significant difference was observed when comparing the overall survival in these three Lund scores groups. Using independent-samples Mann–Whitney U-test to analyze L–M score differences between groups of other prognostic factors, and L–M score difference was independent of other prognostic factors—including AJCC stage, tumor status, nodal status, metastasis, and treatment response.

## 4. Discussion

Rhinosinusitis is commonly seen in patients with NPC before or after treatment. In our study, MRI better detected the difference between tumor invasion of the sinus and sinus mucosal disease [18]. L–M scores correlated with the clinical symptoms in a previous study [16]. A high incidence rate (45.2%) of rhinosinusitis was reported in patients with NPC before treatment [6]. In our study, 34.4% of patients with NPC presented with rhinosinusitis compared with previous studies [19,20]. The difference in rhinosinusitis incidence between our study and previous research was the diagnostic criteria of rhinosinusitis. Our study diagnosed rhinosinusitis based on L–M scores ≥ 4. In a study by Su et al., the diagnosis was based on radiological thickening of the sinus mucosa or fluid retention in the sinuses [6]. Moreover, mild sinus mucosal disease could be misdiagnosed as rhinosinusitis, falsely increasing its incidence. The reason for rhinosinusitis in patients with NPC could be evaluated macroscopically and microscopically. Macroscopically, the sinus drainage obstruction by the primary tumor, sinus wall invasion, and alteration of the resident microbiome in naso-sinus mucosa accounted for the high rhinosinusitis incidence in patients with NPC before treatment. Microscopically, the NPC cells build an inflammatory environment to gain a carcinogenic effect. In Epstein–Barr virus (EBV)-infected nasopharyngeal epithelial cells, the responses to interleukin (IL)-26-induced signal transducer and activator of transcription 3 (STAT3) activation were enhanced to promote tumor progression. Through NF-κB, the STAT3 pathway, and the local production of proinflammatory cytokines, NPC cells could play a role in maintaining and amplifying the local inflammation process [21]. In a single institutional study, approximately 73% of patients with NPC receiving combined chemotherapy and 2D-RT were diagnosed with rhinosinusitis, the most common complication [22]. Decreased mucociliary function, desiccated secretion, and impaired excretion led to sinus secretion retention. Sinus structural changes caused by mucosal edema, choana atresia, nasal turbinate hypertrophy, and the sinus opening stenosis and adhesion could also aggravate the sinus mucosa inflammation. Two-dimensional conventional RT had more sinus mucosal damage than IMRT [23,24]. Patients with NPC receiving IMRT had less sinusitis than conventional RT [6,25,26]. Post-radiation sinusitis incidence in patients with NPC receiving IMRT was 45.2–47% [6,12] within one year after treatment. In long-term follow-up, the rate of post-radiation sinus mucosal change peaked at 3–6 months and stabilized within a year [19,27]. The longest follow-up in these studies was up to five years after RT. PSRT incidence dropped to 16.7% in the fifth year after treatment [28].

Studies suggest that sinus tract inflammation could be a precursor of NPC [9,10,11]. Moreover, local inflammation could induce the granulocyte cells to secrete cytokines to eliminate the microbes. Increased genotoxic agents produced by inflammatory cells could make the epithelial cells and connective tissue more vulnerable to cellular DNA damage, especially during long-term exposure [29]. EBV, a well-known risk NPC factor, could activate inflammatory cytokines pathways that promote cellular growth and proliferation. Activation of Janus kinase (JAK), signal transducer, and STAT3 pathway promotes tumor growth, angiogenesis, and survival. STAT3 activation was mainly mediated by IL-6, a major inflammatory cytokine [30,31]. In EBV-infected naso-epithelial cells (NPE), the IL-6/STAT3 pathway facilitated the malignant transformation and its aggressiveness. EBV-encoded RNAs (EBERs) could elicit a strong inflammatory response in NPC through toll-like receptor 3 (TLR3). Tumor necrosis factor-α overproduction mediated by EBER/TLR3 pathway created a pro-tumorigenic microenvironment for tumor growth. Inflammation-to-carcinogenesis circuit endowed NPC development and invasiveness.

Huang et al. reported that PSRS had a negative predictive role in the local recurrence of NPC [12]. PSRS was also an important prognostic predictor of DFS, freedom from local failure (FFLF), and freedom from distant failure (FFDF). However, PSRS did not correlate with overall survival compared to our study [12]. In our study, the prognostic factors of the overall survival were PSRS and Lund score difference. As opposed to the study by Huang et al., the incidence of PRRS and PSRS in our study was less. Different diagnostic criteria used in evaluating radiological examinations could explain it. We set the diagnostic criteria of rhinosinusitis as L–M scores ≥ 4. Therefore, single sinus disease and mild sinus mucosal inflammation were ruled out. Rhinosinusitis without clinical symptomatic and nasal endoscopic examination was easily regarded as a minimal disease [18]. When we adopted the diagnostic criteria used by Huang et al., the rates of PRRS and PSRS were raised from 34.4% and 36.7% to 62.2% and 72.2%, respectively. The change in diagnostic criteria also made the significant correlation between PSRS and overall survival insignificant. This is the first study to propose that the L–M scores difference was a prognostic factor of NPC. Therefore, patients with NPC with worsening L–M scores possessed poor prognoses. We hypothesized that the tumor status and clinical staging did not affect L–M scores difference, which the multivariate analysis proved. Therefore, L–M score difference < 0 was an independent prognostic factor. The suggestedreasons L–M scores played a prognostic role in NPC were:Poor treatment response would lead to partial tumor regression.Impaired sinus drainage destroys the bony sinus wall, or tumor sinus mucosa invasion would make the sinusitis persist.NPC cells created a proinflammatory microenvironment, putting the nasal cavity and nasopharynx under inflammatory status. Sinus drainage pathway and sinus wall and mucosa restoration indicate substantial tumor volume shrinkage after radiation therapy.

A robust ostiomeatal complex, a stronger immune system, and a healthier local environment reduced the probability of tumor recurrence. Therefore, we recommend that clinicians routinely assess inflammatory rhinosinusitis in patients with NPC during follow-up.

There was a positive trend that the change of severity of sinusitis in NPC patients after treatment was associated with NPC overall survival, although L–M difference < 0 was the only significant factor statistically. The poor prognosis of NPC is multifactorial. These factors may have partly contributed to the rhinosinusitis, but the evidence was limited. Further studies on the molecular aspects are needed to determine the relationship between rhinosinusitis and NPC survival [3].

This study had limitations. First, it was a retrospective, case-series study with small sample size. Prospectively, a case-control study is required to verify our findings further. Second, participants with advanced NPC accounted for two-thirds of our cases, worsening overall survival. Third, L–M scores difference may not predict recurrence probability in the long-term as rhinosinusitis may stabilize after a two-year follow-up.

## 5. Conclusions

Our study revealed that PSRS and worsening L–M score difference played independent prognostic roles in NPC survival. Furthermore, the L–M score system could help clinicians to evaluate rhinosinusitis severity. However, further cohort studies with larger sample sizes are needed to investigate the relationship between rhinosinusitis and NPC prognosis.

## Figures and Tables

**Figure 1 biomedicines-10-03235-f001:**
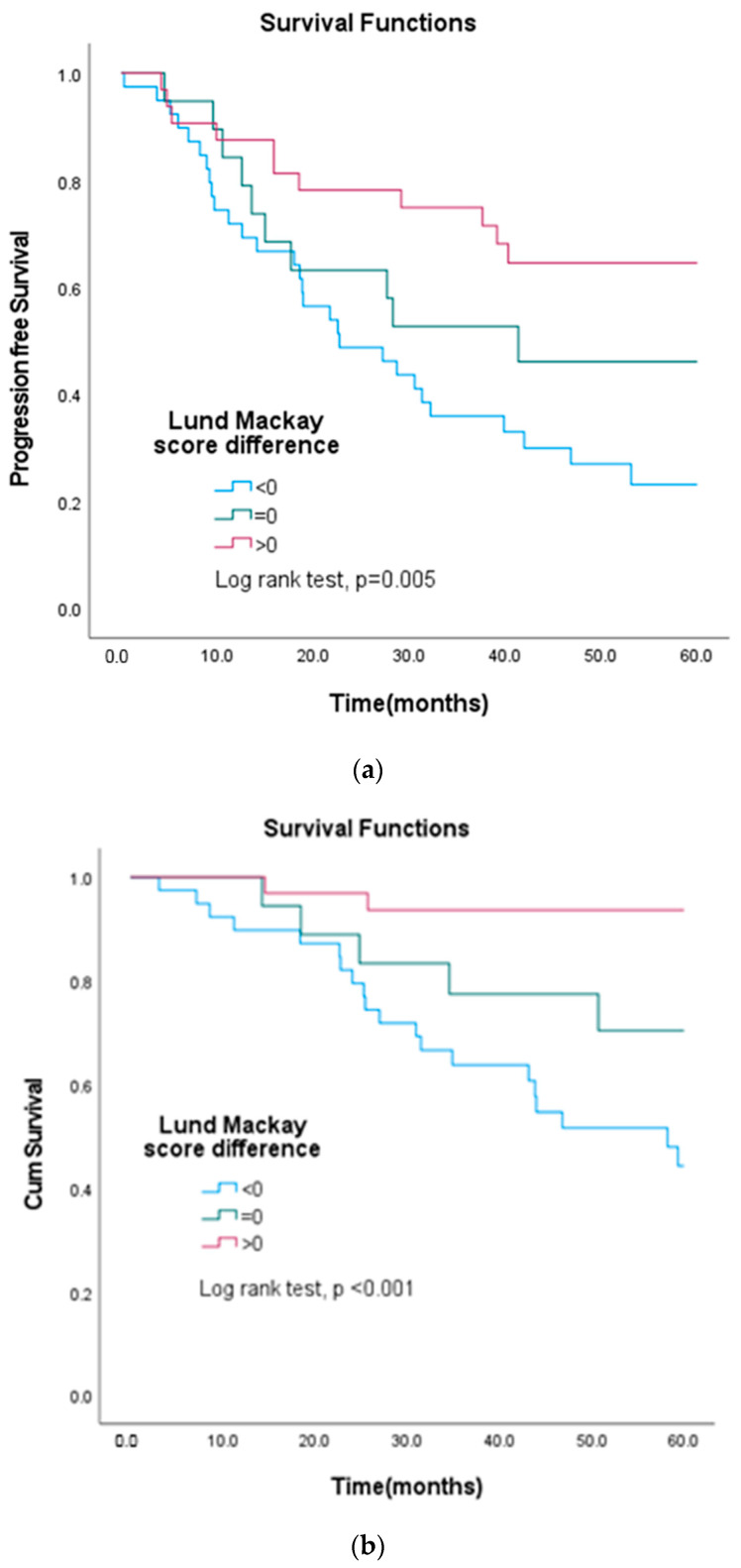
(**a**) The log-rank test of L–M scores difference < 0 had the worst progression-free survival (PFS) compared to L–M score difference > 0 (*p* < 0.001). (**b**) The log-rank test of L–M scores difference < 0 had the worst overall survival compared to L–M score difference > 0 (*p* < 0.001).

**Table 1 biomedicines-10-03235-t001:** Patient Characteristics.

Variables	NPC (N = 90) n (%)
**Sex**	
Female	27 (30.0)
Male	63 (70.0)
**Mean age, y (±SD)**	55.3 ± 12.1
**Age group**	
<45	15 (16.7)
45–64	58 (64.4)
≥65	17 (18.9)
Histology (WHO type)	
Type I	1 (1.1)
Type II	69 (76.7)
Type III	20 (22.2)
**AJCC staging**	
I	9 (10.0)
II	17 (18.9)
III	22 (24.4)
IVa	29 (32.2)
IVb	8 (8.9)
IVc	5 (5.6)
**Treatment**	
RT	12 (13.3)
CCRT	66 (73.3)
NACCRT	12 (13.3)
**Treatment response**	
CR	71 (78.9)
PR	12 (13.3)
Residual	7 (7.8)
**Recurrence**	
None	40 (44.4)
Local	20 (22.2)
Regional	10 (11.1)
Distant	20 (22.2)
**PRCRS**	31 (34.4)
L–M total score	7.7 ± 4.9
**PSCRS**	33 (36.7)
L–M total score	8.1 ± 4.7
**L–M score Txdifferent**	
All	−0.5 ± 4.4
PRCRS	2.0 ± 4.7
PSCRS	−2.8 ± 5.9
Follow-up time (months)	44.9 ± 16.8
Total death	27 (30.0)

AJCC: American Joint Committee on Cancer; RT: radiation therapy; CCRT: concurrent chemoradiotherapy; NACCRT: neoadjuvant chemotherapy following CCRT; CR: complete response; PR: partial response; Residual: stable/progression; PRRS: pre-treatment rhinosinusitis; PSRS: post-treatment rhinosinusitis; L–M score: Lund–Mackay score.

**Table 2 biomedicines-10-03235-t002:** Hazard ratio and 95% confidence interval of death.

Variable	Crude			Adjusted		
	HR	(95% CI)	*p*-Value	HR	(95% CI)	*p*-Value
Sex (male vs. female)	0.63	0.26–1.57	0.324			
Age (every one year)	1.03	0.99–1.06	0.084			
**AJCC staging**						
I	Ref.					
II	1.47	0.15–14.11	0.740			
III	2.88	0.35–23.99	0.327			
IV	5.08	0.67–38.35	0.115			
**Histology (WHO type)**						
Type I and II	Ref.					
Type III	1.06	0.40–2.82	0.902			
**Recurrence (yes vs. no)**	79.1	3.49–1795.82	0.006			
**Tumor**						
1	Ref.			Ref.		
2	1.70	0.49–5.86	0.403	1.05	0.30–3.72	0.943
3	1.29	0.31–5.41	0.728	0.96	0.23–4.07	0.953
4	3.13	1.12–8.74	0.029	2.47	0.85–7.16	0.097
**Node**						
0	Ref.					
1	2.02	0.55–7.48	0.291			
2	3.08	0.84–11.24	0.089			
3	2.78	0.66–11.72	0.164			
**Metastasis (yes vs. no)**	2.85	0.85–9.51	0.089			
**PRRS (yes vs. no)**	0.98	0.44–2.18	0.955			
**PSRS (yes vs. no)**	3.51	1.60–7.68	0.002	1.76	0.72–4.30	0.214
**L–M score Txdifferent group**						
Positive	Ref.			Ref.		
Zero	4.68	0.91–24.13	0.065	5.09	0.98–26.35	0.053
Negative	10.09	2.36–43.18	0.002	8.41	1.86–38.14	0.006
Adjusted: tumor and PSRS						

AJCC: American Joint Committee on Cancer; PRRS: pre-treatment rhinosinusitis; PSRS: post-treatment rhinosinusitis; L–M score: Lund–Mackay score; CI: confidence interval; HR: hazard ratio.

**Table 3 biomedicines-10-03235-t003:** The association between different variables and Lund–Mackay score difference.

Variables	Number of Patients	Lund–Mackay Score Difference Median (Q1, Q3)	*p*-Value
**Tumor**			0.181
1	29	0 (−2.2)	
2	16	−1 (−4.0)	
3	13	0 (−2.1)	
4	32	0 (−3.2)	
**Node**			0.714
0	18	0 (−2.0)	
1	30	0 (−4.3)	
2	27	0 (−2.1)	
3	15	0 (−2.2)	
**Metastasis**			0.210 *
0	85	0 (−2.2)	
1	5	−5 (−6.0)	
**AJCC staging**			0.531
I	9	0 (−2.0)	
II	17	0 (−4.1)	
III	22	0 (−2.3)	
IV	42	0 (−3.2)	
**Treatment**			0.641
RT	12	0 (−2.0)	
CCRT	66	0 (−3.2)	
NACCRT	12	1 (−2.2)	

* Independent-samples Mann–Whitney U-test; RT: radiation therapy; CCRT: concurrent chemoradiotherapy; NACCRT: neoadjuvant chemotherapy following CCRT; independent-samples Kruskal–Wallis Test; AJCC: American Joint Committee on Cancer.

## Data Availability

The datasets generated during and/or analyzed during the current study are available from the corresponding author on reasonable request.

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
