# Peer review of "Worsening Rhinosinusitis as a Prognostic Factor for Patients with Nasopharyngeal Carcinoma: A Retrospective Study"

_biomedicines, 2022, doi:10.3390/biomedicines10123235_

Round 1
Reviewer 1 Report
Dear Authors. thank you for your paper. I've found it interesting. The study is well described and data have been well reported herein. Although the topic is widely discussed in litarature, I think that your paper may add something to the current scientific literature. I hope that the paper topic could fit with the special issue one. Suggested minor text editing. Thank you again for your paper
Author Response
Point 1:
Dear Authors. thank you for your paper. I've found it interesting. The study is well described and data have been well reported herein. Although the topic is widely discussed in litarature, I think that your paper may add something to the current scientific literature. I hope that the paper topic could fit with the special issue one. Suggested minor text editing. Thank you again for your paper
Response #1:
Thank you so much for the acknowledgement. As suggested, the manuscript has been edited for the minor issues.

Reviewer 2 Report
The authors demonstrated that worsening rhinosinusitis was a prognostic factor for patients with NPC. Clinicians should take it as a warning sign of recurrence in the routine follow-up. Several main points as below need to be note to improve the quality of this manuscript.
1) In “Abstract” section, the authors concluded that “Clinicians should take it as a warning sign of recurrence in the routine follow-up”, However, no clinical data about “Recurrence” was analyzed in the study.
2) In “Introduction” section, as to “NPC is a distinct head and neck cancer entity due to its unique pathophysiology and treatment modalities”, more details should be provided. For example, as we know, NPC tends to have lymph nodes metastasis when diagnosed and develop local recurrence and distant metastasis after therapy. The causes are complex. An interesting study recently propose the ecology of NPC, and NPC progression should be bidirectional that accounts for the recurrence and distant metastasis. (Nasopharyngeal Carcinoma Ecology Theory: Cancer as Multidimensional Spatiotemporal “Unity of Ecology and Evolution” Pathological Ecosystem. Preprints. 2022; 2022100226. Please check, https://www.preprints.org/manuscript/202210.0226/v2). These views might be better updated.
3) In “Materials and Methods” section, â‘ Please provide the approved number by the institutional review board. â‘¡ The reference about the L-M score proposed should be added.
4) In “Results” section, â‘ Table 1. Patient Characteristics, please add the factors of tumor histology, T, N, M and recurrence. â‘¡ Table 2, add variable “Tumor histology” and “Recurrence”.
5) Why no significant association was observed between the Lund score and any of clinical characteristics, whereas patients with L-M score difference <0 had the worst overall survival, others had better?
6) In Figure 1, how about the impact of L-M score difference on progression-free survival (PFS) ?
Author Response
Point 1:
In “Abstract” section, the authors concluded that “Clinicians should take it as a warning sign of recurrence in the routine follow-up”, However, no clinical data about “Recurrence” was analyzed in the study.
Response #1:
Thank you for your careful observation. We have revised the text as follows:
P1 L22-23
“Therefore, worsening rhinosinusitis was a prognostic factor for patients with NPC. Clinicians should consider NPC as a warning sign of poor prognosis during routine follow-ups.”
Point 2:
In “Introduction” section, as to “NPC is a distinct head and neck cancer entity due to its unique pathophysiology and treatment modalities”, more details should be provided. For example, as we know, NPC tends to have lymph nodes metastasis when diagnosed and develop local recurrence and distant metastasis after therapy. The causes are complex. An interesting study recently propose the ecology of NPC, and NPC progression should be bidirectional that accounts for the recurrence and distant metastasis. (Nasopharyngeal Carcinoma Ecology Theory: Cancer as Multidimensional Spatiotemporal “Unity of Ecology and Evolution” Pathological Ecosystem. Preprints. 2022; 2022100226. Please check, https://www.preprints.org/manuscript/202210.0226/v2). These views might be better updated.
Response #2:
Thank you for providing these insights. We have added the reference and edited our manuscript as follows:
P1 L30-33
“NPC is a distinct head and neck cancer entity due to its multifactorial pathogenesis, including genetic and environmental factors and pathogenic infections, like Epstein-Barr virus (EBV). A theory of “NPC ecology” has been proposed to build a new comprehensive framework of NPC tumorigenesis and tumor progression.”
Reference:
- Luo W. Nasopharyngeal Carcinoma Ecology Theory: Cancer as Multidimensional Spatiotemporal “Unity of Ecology and Evolution” Pathological Ecosystem. 2022.
Point 3:
In “Materials and Methods” section, â‘ Please provide the approved number by the institutional review board. â‘¡ The reference about the L-M score proposed should be added.
Response #3:
Thank you for the suggestions. We have added the approval number and the reference.
P2 L68-70
This study was approved by the Institutional Review Board of the Taichung Tzu Chi Hospital (protocol code: REC111-65).
P3 L117-119
“The L–M score, proposed by Valerie J. Lund and Ian S. Mackay [16], recorded changes in the frontal, anterior ethmoid, posterior ethmoid, and sphenoid sinuses and ostiomeatal complex.”
Reference:
- Lund, V.J.; Mackay, I.S. Staging in rhinosinusitis. Rhinology 1993, 31, 183–184.
Point 4:
In “Results” section, â‘ Table 1. Patient Characteristics, please add the factors of tumor histology, T, N, M and recurrence. â‘¡ Table 2, add variable “Tumor histology” and “Recurrence”.
Response #4:
Thank you for the valuable comment.
- We have added the tumor histology, TNM score, and WHO histology in Table I.
- We have added the tumor histology and recurrence details in Table 2.
Table I
Histology (WHO type)
Type I 1 (1.1)
Type II 69 (76.7)
Type III 20 (22.2)
Recurrence
None 40 (44.4)
Local 20 (22.2)
Regional 10 (11.1)
Distant 20 (22.2)
Table II
Histology (WHO type)
Type I and II Ref.
Type III 1.06 0.40-2.82 0.902
Recurrence (yes vs. no) 79.1 3.49-1795.82 0.006
Based on the WHO pathological classification, there was 1 patient (1.1%) in type I, 69 patients (76.7%) in type II, and 20 patients (22%) in type III.
P149-151
Twenty patients (22.2%) developed local recurrence, and 10 patients (11.1%) developed regional recurrence, and 20 patients (22.2%) developed distant metastasis after treatment.
Point 5:
Why no significant association was observed between the Lund score and any of clinical characteristics, whereas patients with L-M score difference <0 had the worst overall survival, others had better?
Response #5:
Thank you for the comment.
There was a positive trend that the change of severity of sinusitis in NPC patients after treatment was associated with NPC survival, although L-M difference<0 was the only significant factor statistically. The poor prognosis of NPC was multifactorial. These factors may have partly contributed to rhinosinusitis, but the evidence was limited. Further studies on the molecular aspects are needed to determine the relationship between rhinosinusitis and NPC survival.
Point 6
In Figure 1, how about the impact of L-M score difference on progression-free survival (PFS) ?
Response #6:
Thank you for the valuable suggestion.
As suggested, we have added the Kaplein-Meier curve about PFS vs. L-M score difference in Figure 1b (given below). The conclusion was L-M difference<0 was a significant prognostic factor in PFS.
P2 L193-197
The Kaplan–Meir survival revealed that the three L–M score groups had different pro-gression free survival (PFS) and overall survival (OS) scores.
Patients with L–M score difference <0 had the worst progression free survival and overall survival scores (Figure 1a, b).

Round 2
Reviewer 2 Report
The authors have answered all of the concerns. No other questions.